# Cryo-EM reveals the conformational epitope of human monoclonal antibody PAM1.4 broadly reacting with polymorphic malarial protein VAR2CSA

Sai Sundar Rajan Raghavan[1], Robert Dagil[1], Mary Lopez-Perez[1], Julian Conrad[2], Maria Rosaria Bassi[1], Maria del Pilar Quintana[1], Swati Choudhary[1], Tobias Gustavsson[1], Yong Wang[3], Pontus Gourdon[4,5], Michael Fokuo Ofori[6], Sebastian Boje Christensen[1], Daniel Thomas Remias Minja[7], Christentze Schmiegelow[1], Morten Agertoug Nielsen[1], Lea Barfod[1], Lars Hviid[1], Ali Salanti[1], Thomas Lavstsen[1]*, Kaituo Wang[4]*

1 Centre for Medical Parasitology at Department for Immunology and Microbiology, Faculty of Health and Medical Sciences, University of Copenhagen, and Department of Infectious Diseases, Rigshospitalet, Copenhagen, Denmark, 2 Swedish National Cryo-EM Facility, Science for Life Laboratories, Solna, Sweden, 3 Joint Research Centre for Engineering Biology, Zhejiang University-University of Edinburgh Institute, College of Life Sciences, Zhejiang University, Hangzhou, Zhejiang, China, 4 Department of Biomedical Sciences, University of Copenhagen, Copenhagen, Denmark, 5 Department of Experimental Medical Science, Lund University, Lund, Sweden, 6 Department of Immunology, Noguchi Memorial Institute for Medical Research, University of Ghana, Legon, Ghana, 7 National Institute for Medical Research, Tanga Centre, Tanga, Tanzania

* thomasl@sund.ku.dk (TL); kaituo@sund.ku.dk (KW)

**Data Availability Statement:** All data related to the findings in the article are contained in the manuscript and Supporting information. The

## Abstract

Malaria during pregnancy is a major global health problem caused by infection with *Plasmodium falciparum* parasites. Severe effects arise from the accumulation of infected erythrocytes in the placenta. Here, erythrocytes infected by late blood-stage parasites adhere to placental chondroitin sulphate A (CS) via VAR2CSA-type *P. falciparum* erythrocyte membrane protein 1 (PfEMP1) adhesion proteins. Immunity to placental malaria is acquired through exposure and mediated through antibodies to VAR2CSA. Through evolution, the VAR2CSA proteins have diversified in sequence to escape immune recognition but retained their overall macromolecular structure to maintain CS binding affinity. This structural conservation may also have allowed development of broadly reactive antibodies to VAR2CSA in immune women. Here we show the negative stain and cryo-EM structure of the only known broadly reactive human monoclonal antibody, PAM1.4, in complex with VAR2CSA. The data shows how PAM1.4's broad VAR2CSA reactivity is achieved through interactions with multiple conserved residues of different sub-domains forming conformational epitope distant from the CS binding site on the VAR2CSA core structure. Thus, while PAM1.4 may represent a class of antibodies mediating placental malaria immunity by inducing phagocytosis or NK cell-mediated cytotoxicity, it is likely that broadly CS binding-inhibitory antibodies target other epitopes at the CS binding site. Insights on both types of broadly reactive monoclonal antibodies may aid the development of a vaccine against placental malaria.

atomic coordinates and electron-microscopy data have been deposited in the RCSB Protein Data Bank and in the Electron Microscopy Data Bank under the following entries: VAR2CSA complex with PAM1.4 (PDB:7Z12; EMDB ID: EMD-14438) and APO VAR2CSA (PDB:7Z1H, EMDB ID: EMD-14446).

**Funding:** SSRR, KT and TL are funded by The Lundbeck Foundation (R344-2020-934) and the Independent Research Fund Denmark (9039-00285A). LH, MLP and MPQ are supported by Independent Research Fund Denmark (0134-00123B), Danida (17-02 KU), NIH (R21AI164147-01), and EU (101028915). LB and MRB are funded by Novo Nordisk Foundation (NNF170C0026778). AS, RD, TG, SC are funded by Novo Nordisk Foundation grants NNF21OC0068192 and NNF22OC0076055. SC, TG, MAN, AS and TL are funded by a Semper Ardens Grant from Carlsberg Foundation. MAN is supported by the Global Health Innovative Technology Fund (G2020-214) and the European Union Advance-vac4PM, Grant number: 101057882 (Views and opinions expressed are however those of the authors only and do not necessarily reflect those of the European Union or the European Health and Digital Executive Agency. Neither the European Union nor the granting authority can be held responsible for them funded by the European Union). CS is funded by Independent Research Fund Denmark (1030-00371B). The FOETALforNCD study was funded by The Danish Council for Strategic Research (1309-00003B). The funders had no role in study design, data collection, decision to publish or preparation of the manuscript.

**Competing interests:** We have read the journal's policy and the authors of this manuscript have the following competing interests: AS, and MAN are listed as co-inventors on a patent family covering the use of VAR2CSA to target and diagnose cancer. AS and is listed as co-inventor on a patent on using VAR2CSA as a prophylactic malaria vaccine during pregnancy. The other authors have no conflicts of interest.

## Author summary

Placental malaria remains a major global health problem. The disease is caused by the accumulation of malaria parasite-infected red blood cells in the placenta. The parasites export members of the polymorphic VAR2CSA protein family onto the red blood cell surface to bind placental chondroitin sulphate A (CS). The VAR2CSA protein family have diversified in sequence to avoid immune recognition, but recent CryoEM structures of VAR2CSA shows that all variants fold into a large, dense and stable core comprised of multiple interwoven "DBL" domains, and a flexible two-DBL domain tail.

Immunity to placental malaria is acquired through exposure and thought to be mediated by broadly reactive antibodies acting through neutralizing CS binding or opsonizing infected red blood cells for cell-mediated killing. Here, we used negative stain and cryo-EM to resolve the first molecular structure of a broadly reactive antibody (PAM1.4) against VAR2CSA. We find that the antibody binds a highly conserved conformational epitope on the VAR2CSA core structure, distant from previously determined CS binding sites. This aligns with new data showing negligible PAM1.4 neutralization of parasite binding to CSA. As a well-characterized non-neutralizing monoclonal antibody, PAM1.4 and Fc-modifications thereof, can now be used for reference and benchmarking in future studies aiming to map, identify and qualify functional antibodies against VAR2CSA in the continued quest for vaccines or therapeutics against placental malaria.

## Introduction

Each year, over 10 million women experience infection with *Plasmodium falciparum* malaria parasites during pregnancy. Placental malaria imposes significant threats to both mother and child health [1], with risk of maternal anaemia, hypertension, stillbirth, premature delivery and low birth weight being particularly high in first pregnancies [2, 3]. Placental malaria is caused by extensive accumulation of *P. falciparum*-infected erythrocytes in the placenta [2]. *P. falciparum* parasites sequester from the blood circulation to escape splenic destruction of infected erythrocytes [4]. The tissue sequestration of infected erythrocytes is mediated by members of the *P. falciparum* Erythrocyte Membrane Protein 1 (PfEMP1) family [5] which has expanded to confer several mutually exclusive receptor binding phenotypes and diversified in sequence to avoid immune recognition (antigenic diversity) [6]. In non-pregnant women, infected erythrocyte sequestration is mediated by PfEMP1 that can bind to endothelial receptors such as Endothelial Protein C Receptor (EPCR), Intracellular Adhesion Molecule 1 (ICAM-1) or CD36, which has been associated with severe malaria in immunologically naïve hosts (children) and asymptomatic infections in semi-immune hosts, respectively [7]. In pregnancy, however, the placenta provides the parasite with an additional niche that enables parasite adhesion to CS via VAR2CSA-type PfEMP1 [8, 9].

VAR2CSA-type PfEMP1 are ~310 kDa large, multi-domain proteins embedded in the erythrocyte membrane. They are composed of a short and conserved intracellular domain, a transmembrane domain and a large, polymorphic ectodomain. Each parasite genome typically encodes one or two VAR2CSA variants with ectodomains differing in sequence within and among genomes [6, 10].

The VAR2CSA ecto-domain is composed of a short N-terminal segment, six Duffy Binding-Like (DBL) domains (DBL1-DBL6), and three inter-domain regions (ID1-3). All are unique to VAR2CSA-type PfEMP1. Extensive efforts to resolve the molecular basis of

VAR2CSA binding (reviewed in [11]), established the CS binding site to be contained within the region spanning ID1-DBL2-ID2. These studies recently culminated in the molecular resolution data of the interaction of the full ectodomain VAR2CSA protein variants with CS [12–14]. These studies confirmed speculations that all VAR2CSA variants adopt a similar fold with the ID1-DBL2-ID2-DBL3-DBL4-ID3 region forming a compact, stable core containing a CS binding pore. The core is decorated with highly diverse flexible loops and surface residues.

The development of immunity against placental malaria correlates with acquisition of antibodies to VAR2CSA, and significant protection is gained after first pregnancies [3, 15, 16]. Immunity is likely mediated by antibodies inhibiting VAR2CSA binding to placental CSA [17] and opsonizing VAR2CSA+ infected erythrocytes for cell-mediated destruction [18–20]. Both variant-specific and broadly reactive antibodies are acquired by placental malaria-immune individuals [21], with the latter possibly relying on conformational epitopes involving multiple domains. The exact epitopes recognized by broadly cross-reactive antibodies have not been identified, which has frustrated attempts to develop VAR2CSA-based vaccines against PM. Vaccination of rodents with truncated VAR2CSA proteins can elicit antibodies that inhibit diverse VAR2CSA variants like antibodies from naturally VAR2CSA-exposed women [22, 23]. However, these vaccine candidates induced only low titres of antibodies and very limited cross-reactivity in human phase I clinical trials [11, 24, 25].

To date, only eight human monoclonal anti-VAR2CSA antibodies (mAbs) have been identified and characterized. Of these, five exhibit reactivity with several (but not all) sequence diverse natively expressed VAR2CSA variants, whereas one, PAM1.4, appears to recognize all (or at least most) native VAR2CSA variants [26–28]. PAM1.4 can efficiently opsonize infected erythrocytes for Fc receptor-dependent killing *in vitro* [18, 29], whereas recent studies revoking early data have led to the consensus notion that PAM1.4 is unable to inhibit infected erythrocyte adhesion to CS [30].

Building on recent cryogenic Electron Microscopy (cryo-EM) protocols for structural studies of VAR2CSA, we analysed PAM1.4 Fab complexed with full-length VAR2CSA (FL VAR2CSA). We found that PAM1.4 Fab binds a highly conserved conformational epitope that involves several of the domains composing the core VAR2CSA structure.

## Results

### Reactivity of PAM1.4 Fab to recombinant and native VAR2CSA

The ability of the PAM1.4 Fab fragment to bind recombinant full-length VAR2CSA and VAR2CSA domains were tested in ELISA with immobilized PAM1.4 Fab fragments. No binding was observed to the minimal CS-binding region proteins (ID1-ID2a and DBL1-ID2a), DBL4 or DBL5, whereas binding to full-length VAR2CSA showed saturation with a calculated $K_D$ of 12 nM (Fig 1A). The previously reported broad cross-reactivity of PAM1.4 with allelic variants of native VAR2CSA was confirmed by flow cytometry and three genotypically distinct parasite lines (IT4/FCR3, NF54 and 7G8) expressing VAR2CSA-type PfEMP1 (Figs 1B and S1). The binding of PAM1.4 Fab to recombinant full-length VAR2CSA was further investigated by negative stain electron microscopy. Comparing the negative stain density maps of APO VAR2CSA (un-complexed VAR2CSA) (Fig 1C and 1D) and PAM1.4 Fab VAR2CSA indicated that PAM1.4 Fab binds to the core structure of VAR2CSA (Fig 1E and 1F). Further model fitting of the previously published cryo-EM structure of VAR2CSA onto the negative stain map of PAM1.4 FabVAR2CSA complex suggested that the PAM1.4 Fab binds a conformational epitope involving multiple domains of the core structure (S2 Fig).

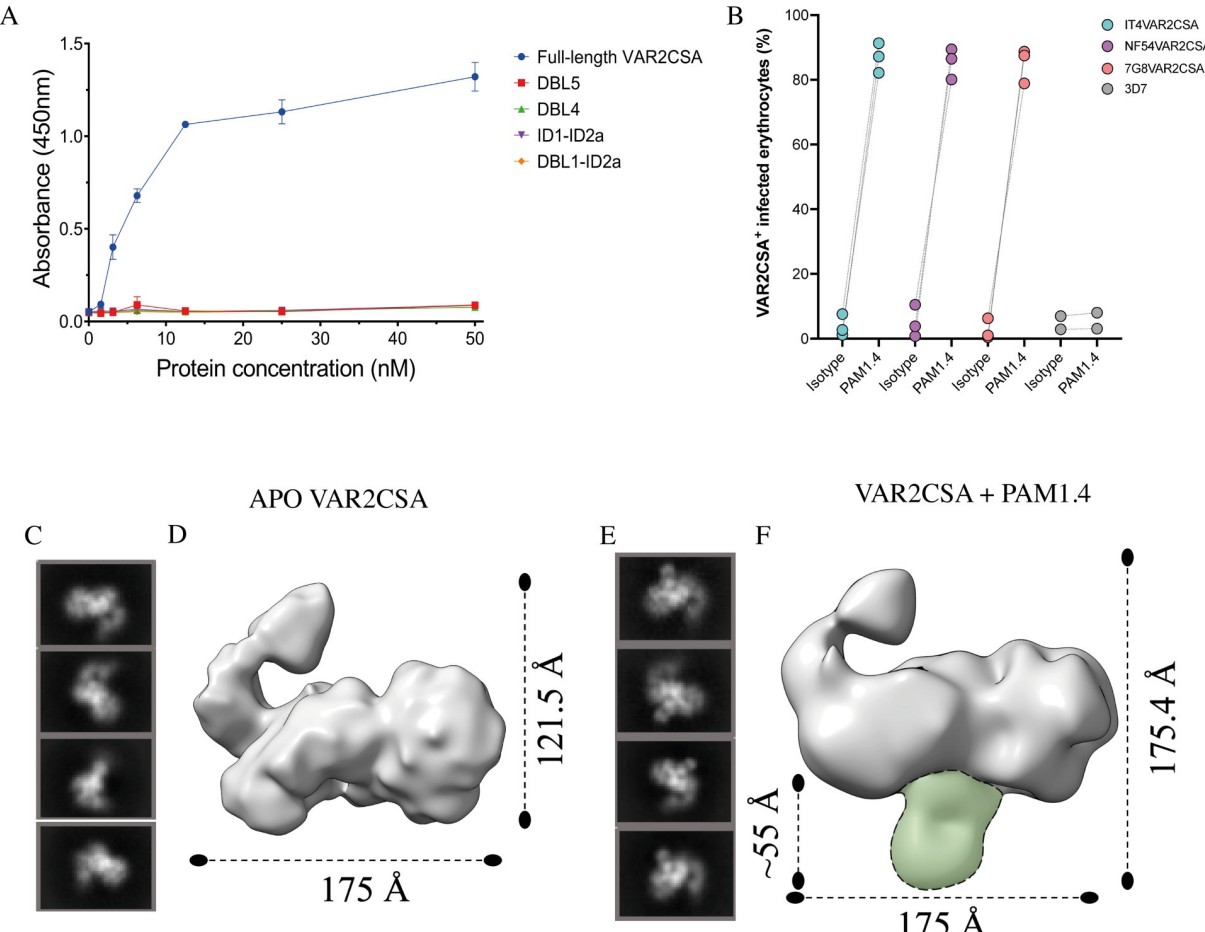

**Fig 1. Reactivity of PAM1.4 to full-length recombinant VAR2CSA and native VAR2CSA.** (a) ELISA analysis showing binding of PAM1.4 Fab to full-length recombinant VAR2CSA and not to single domain constructs ID1-1D2a, DBL1-ID2a, DBL4, DBL5. (b) *P. falciparum* laboratory-adapted clones FCR3, NF54, and 7G8 expressing VAR2CSA on the infected erythrocyte surface and a non-VAR2CSA 3D7 clone were incubated with 10µg/ml PAM1.4 IgG or IgG isotype control and evaluated by flow cytometry as described to confirm reactivity to the native protein. Independent experiments are shown as individual points. (c) Representative of reference free 2D classification of APO VAR2CSA. (d) Refined 3D models after iterative rounds of 2D and 3D classification. (e) 2D class averages of VAR2CSA complexed with PAM1.4 Fab. (f) Refined 3D model of the complex VAR2CSA and PAM1.4 Fab. Extra density corresponding to the Fab is highlighted in green.

## Cryo-EM structure of PAM1.4 Fab complexed with full length FCR3 VAR2CSA

To understand the molecular details of PAM1.4 binding to VAR2CSA, we determined the cryo-EM structure of PAM1.4 Fab in complex with the full ectodomain of VAR2CSA (variant FCR3) (Fig 2) at an overall resolution of 3 Å and with resolution of the binding region reaching 2.5 Å (S3 Fig). The PAM1.4 Fab was found to bind the globular VAR2CSA core structure away from the main surface-exposed CS binding site, which has also been suggested as the main entry point to the core-traversing CS-binding pore (Fig 3A). The antibody bound a concave face of the VAR2CSA core structure with an epitope spanning exposed residues of ID1, DBL2, ID2 and DBL4 (Figs 2 and S4). Comparing the PAM1.4 Fab VAR2CSA complex structure with structural data of APO VAR2CSA (RMSD: 0.909Å) derived here at a resolution of 3.12Å (Figs 3B, S5 and S3), did not indicate major conformational change to the VAR2CSA core structure. These observations explain why PAM1.4 does not bind individual single domains of

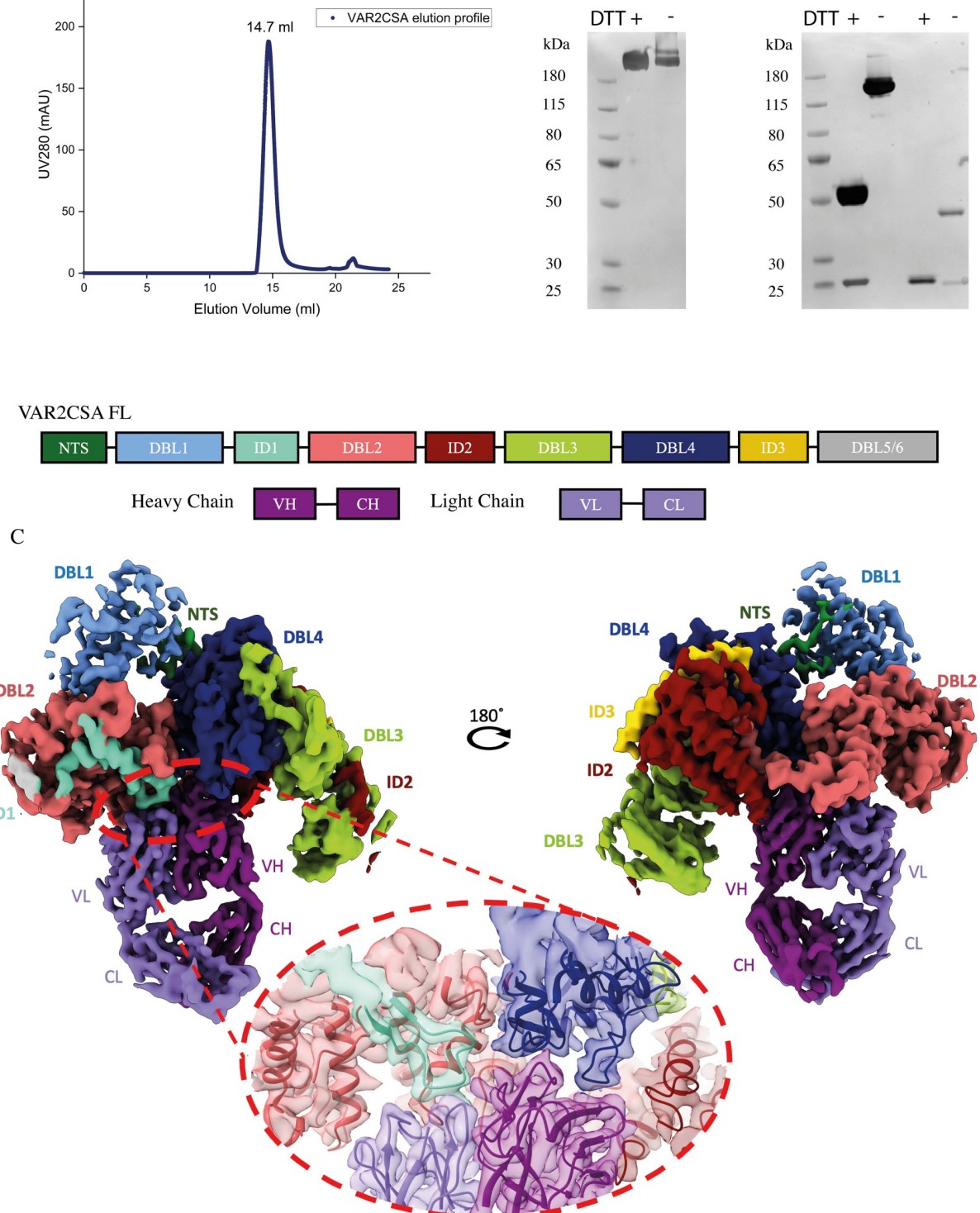

**Fig 2. CryoEM reveals conformational epitope binding of PAM1.4 Fab to VAR2CSA.** (a). Gel filtration profile of VAR2CSA full length in 20mM Tris HCl, 150mM KCl, pH. 7.5 buffer. (b) Silver stained SDS-PAGE reduced/ non-reduced of full-length VAR2CSA (307 kDa), PAM1.4 IgG and PAM1.4 Fab after IgG digestion and Fc removal by reverse protein A purification. (c) Overall architecture of VAR2CSA with PAM1.4 Fab, flipped 180˚. The domains are colored individually. NTS–N-terminal Sequence, DBL- Duffy Binding Like domain, ID- Inter-domain, VH, CH–Variable and Constant Heavy, VL, CL–Variable and Constant Light. The binding region is magnified and highlighted with 50% transparency to show conformational epitope binding.

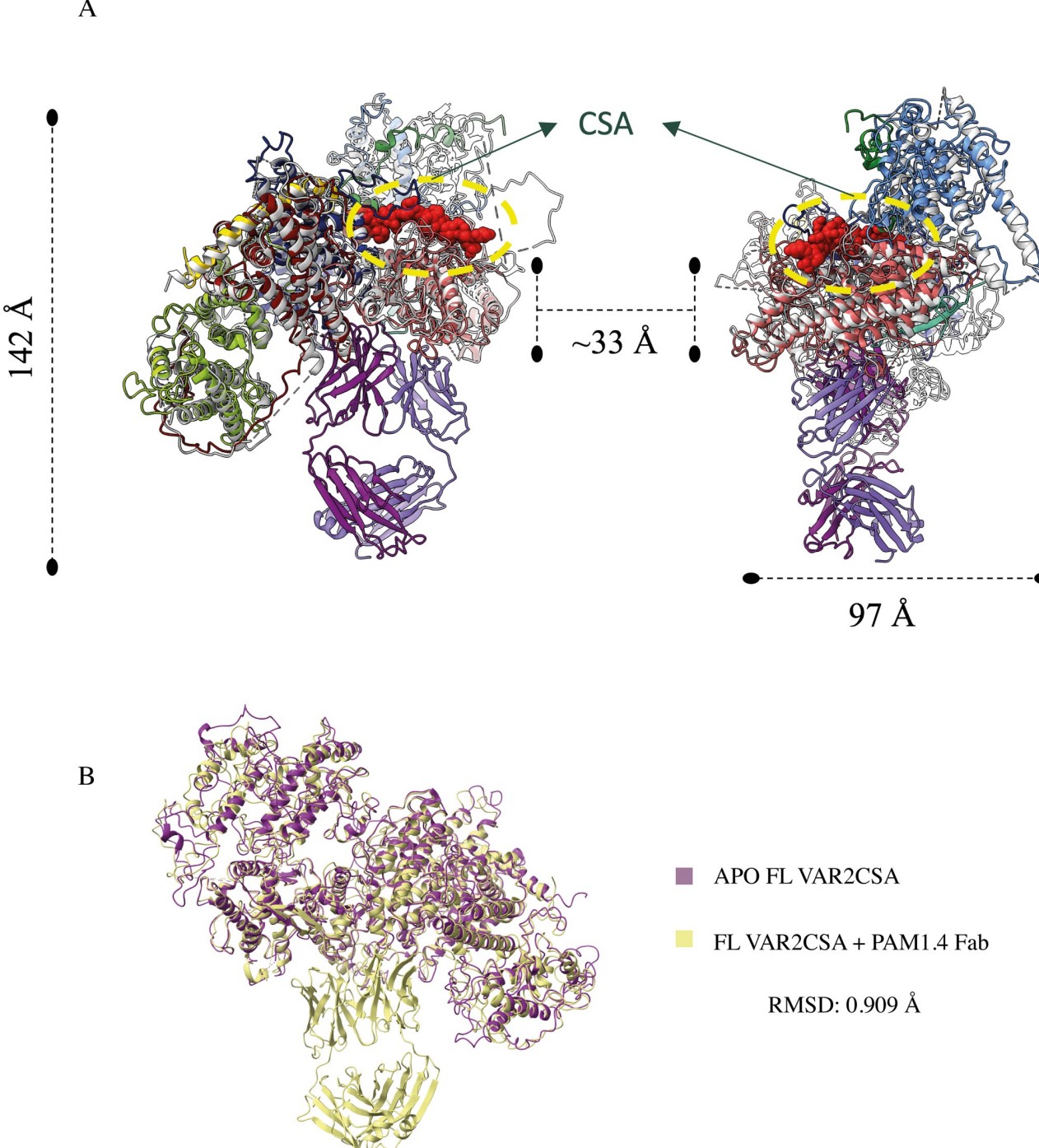

**Fig 3. Overlay of VAR2CSA PAM 1.4 Fab complex structure with VAR2CSA complexed with CSA (PDB ID: 7JGH) and APO VAR2CSA.** (a) Alignment of the models show no major conformational changes between the structures, generating a RMSD of 1.078Å. CSA is shown in red spherical model and the nearest distance between the epitope of CSA and epitope of PAM 1.4 Fab is measured to be ~33Å showing that the CSA binding site is not affected by PAM 1.4 Fab binding. (b) Overlay of APO VAR2CSA (S5 Fig) and VAR2CSA PAM 1.4 Fab complex structure. Alignment of the models shows no major conformational changes on VAR2CSA upon PAM1.4 Fab binding. RMSD: 0.909 Å.

VAR2CSA but binds to full-length VAR2CSA with high affinity, and that predict PAM1.4 is unlikely to inhibit CS binding by VAR2CSA, directly or through induced conformational changes to VAR2CSA. In agreement with this, we did not observe inhibition of VAR2CSA+ infected erythrocyte adhesion to CS, neither by PAM1.4 IgG nor by PAM1.4 Fab (S6 Fig).

## Molecular details of PAM1.4 Fab binding to VAR2CSA

Our data show that PAM1.4 Fab interacts with ID1, DBL2, ID2 and DBL4 of VAR2CSA and that the CDR1, CDR2 and CDR3 regions of both the heavy and light chain of the antibody are involved. The binding is stabilized by hydrogen bonds, electrostatic and hydrophobic interactions.

In ID1, residues K510 and R511 form hydrogen bonds with both light and heavy chain residues of PAM1.4 D47 (CDR-L1), N111 (CDR-L3) and T126 (CDR-H3) (Fig 4). In comparison with the APO VAR2CSA structure, we observe stabilization of ID1 (N521 –K522) loops upon PAM1.4 Fab interaction (S7A Fig).

In DBL2, PAM1.4 Fab binding is mainly stabilized through interactions with residues placed in the loop (L905 –T992) leading to ID2. PAM1.4 residues Y51 (CDR-L1), Y110 (CDR-L3), and T72 (CDR-L2) form hydrogen bonds with VAR2CSA residues D906, D907 and N908, and PAM1.4 R120 (CDR-H3) forms an electrostatic interaction with VAR2CSA D907 (Fig 5). In addition, PAM1.4 Y51 (CDR-H1) and N121 (CDR3-H3) each form a hydrogen bond with side chains of VAR2CSA Y958 and R959, respectively.

In DBL4, PAM1.4 D49 (CDR-H1) forms electrostatic and hydrogen bond with the side chain of R1617 from VAR2CSA (Fig 4). There is also a non-CDR loop of the heavy chain interacting with VAR2CSA DBL4. N93 (IGVH-FRW3) forms a hydrogen bond with G1616 in DBL4 of VAR2CSA.

Finally, PAM1.4 Fab binding to the DBL2 and DBL4 domains is further stabilized by hydrophobic interactions at residues K914-T918 of DBL2 and K1872-E1875 of DBL4. P94 (IGVH-FRW3) forms a hydrophobic contact with I1055 of ID2 loop (S8 Fig).

## PAM1.4 epitope conservation

The degree of conservation of the FCR3 VAR2CSA residues found to directly interact with PAM1.4 was assessed using a recently published sequence alignment of 3737 VAR2CSA variants representing the global diversity of the protein family [13]. Contact residues in ID2 and DBL4 as well as residues L905-N908 and K914-T918 of DBL2 (Figs 4, 5 and S8) were all ~100% conserved. The ID1 residues K510-R511-G512 in FCR3 VAR2CSA were semi-conserved in 76% of VAR2CSA variants (Fig 4). The sequences of four VAR2CSA variants that are recognized by PAM1.4 in their native form (NF54, FCR3, 7G8 and M200101 [27]), represent the observed sequence diversity in position 510–511. However, the K510-R511 are absent from 24% of VAR2CSA variants as they are part of ID1, which segregates VAR2CSA-type PfEMP1 into variants with type 1 and type 2 ID1 regions, respectively [31].

Residues Y958 and R959 in FCR3 DBL2, which were found to interact with PAM1.4 N121 (CDR-H3) and Y51 (CDR-H1), respectively, are the least conserved contact residues in VAR2CSA (Fig 5). PAM1.4 binding to native NF54 and M200101 VAR2CSA, both of which are H958-G959 variants, indicates that PAM1.4 can bind to VAR2CSA independent of Y958 and R959. The adjacent residues, Y960 and K961, are fully conserved among VAR2CSA. These residues homologous to Y958 and R959 are placed close to each other in a highly flexible loop. It is therefore possible that PAM1.4 bind to Y960 and K961 in VAR2CSA variants where Y958 and R959 are absent and replaced by H958 and G959.

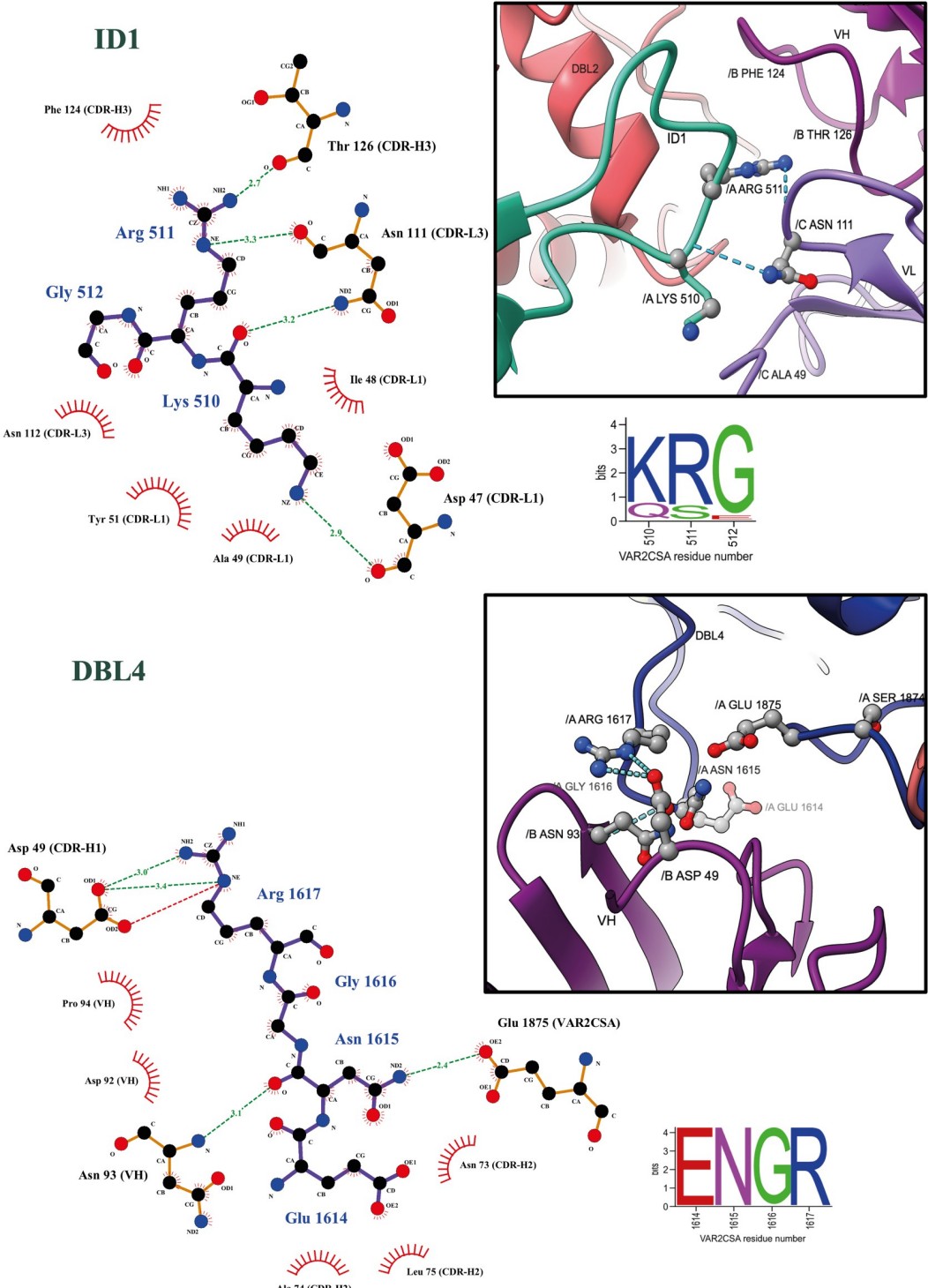

**Fig 4. PAM1.4 Fab interaction analysis to ID1 and DBL4.** Binding plot analysis using Ligplot+ v.2.7. Hydrogen bonds are shown in green dotted lines, electrostatic interactions in red dotted lines and hydrophobic contacts are shown as short spokes radiating from each atom or residue. Sequence conservation of PAM1.4-interacting amino acids in FCR3 VAR2CSA is shown in sequence LOGOs extracted from a previously published sequence alignment LOGO of 3737 VAR2CSA variants found in [13]. FCR3 VAR2CSA sequence positions 510 and 1614 correspond to alignment positions 523 and 1760 respectively. Representation of corresponding Ligplot+ in UCSFChimeraX showing ID1 and DBL4 interactions to Heavy and light chain CDRs of PAM1.4 Fab. Hydrogen bonds are represented in dotted blue lines. Non-CDR residue N93 of heavy chain is shown to form hydrogen bond with N1615 of DBL4 domain in VAR2CSA.

Overall, the sequence and structural conservation of the PAM1.4 epitope explains the previously observed broad cross-reactive nature of PAM1.4 and suggests that PAM1.4 would bind most of the VAR2CSA variants with high affinity.

## PAM1.4 characteristics

The PAM1.4 IGHV and IGLV sequences were analysed to infer the closest related germline sequence, boundaries of framework and complementarity-determining regions (FWRs and CDRs) as well as mutations from nearest germline sequence (S9 Fig). A total of 11 and six amino acid substitutions from nearest germline sequence were found in the IGHV and IGLV framework sequences, respectively. An additional five and two amino acid substitution of CDR1-2 was observed in PAM1.4 IGHV and IGKV, respectively. These amino acid mutations arise from 35 and 14 single nucleotide differences (silent and non-silent), which agrees with previously reported mutation rates for IgG sequences derived from memory B cells [32]. Also, the observed CDR lengths (e.g. VH CDR3 = 16 amino acid residues) are as typically seen for human IgG [33].

## Discussion

The VAR2CSA protein family is considered the main target of antibodies protecting women from adverse events due to *P. falciparum* infection during pregnancy [11, 34]. Only a few human monoclonal VAR2CSA antibodies have been isolated, and only PAM1.4 has shown broad reactivity across the VAR2CSA-type PfEMP1 [29]. The antibody was initially described as both broadly reactive and inhibitory to CS binding [26, 29]. However, recent research by our and other research teams led to the consensus notion that PAM1.4 is cross-reactive but not neutralizing [30]. The functional and structural data presented here support that PAM1.4 is indeed broadly cross-reactive, but that it does not inhibit the interaction between VAR2CSA and CS.

In line with previous work [26] and the ELISA results presented here, our molecular analysis showed that PAM1.4 binds a conformational epitope dependent on the assembly of the core VAR2CSA macromolecular structure. The PAM1.4 epitope is placed in the core structure at a position that is remote from the CSA binding site, at a region exposing closely spaced residues of multiple VAR2CSA domains: ID1, DBL2, ID2 and DBL4. The PAM1.4 interacts through both its heavy and light chains and appears to have undergone affinity maturation at a level normally seen for memory B-cell receptors. Based on sequence conservation of directly interacting VAR2CSA residues, we propose that PAM1.4 can bind all VAR2CSA variants with ID1 type 1 sequences corresponding to three quarters of all VAR2CSA variants. The structure of the ID1 type 2 is unknown and PAM1.4 interaction with *in-silico* predicted structures is unlikely to provide definitive proof of interaction. Thus, further studies are needed to resolve this.

Broadly reactive human monoclonal antibodies to VAR2CSA are likely to be common among women naturally exposed to VAR2CSA. Like PAM1.4, such antibodies may require intact multi-domain constructs to be detected and isolated–and indeed to be induced. In support of this, a recent paper showed that human VAR2CSA (polyclonal) IgG could be depleted from human sera by a single full-length VAR2CSA ectodomain [21] but not by individual VAR2CSA domains [35]. Moreover, affinity purification using a single VAR2CSA domain extracted the majority of cross-reactive inhibitory IgG from pooled serum IgG of malaria-exposed women. Thus, it is possible that broadly neutralizing VAR2CSA antibodies function by direct competition with CS at the CS binding site or sterically inhibiting upon binding near the CS epitopes. CS-binding inhibition is thought to be a key mechanism of acquired

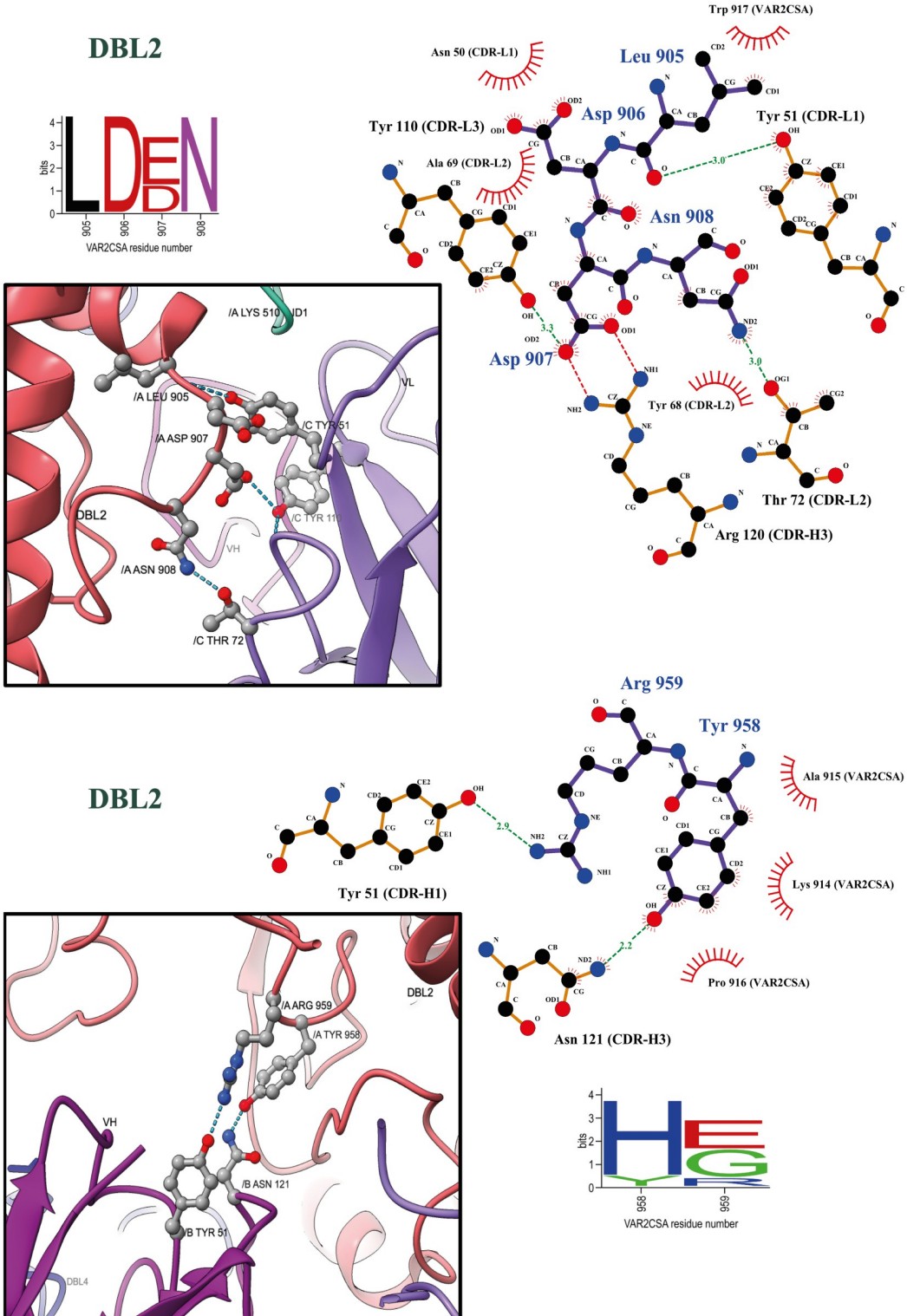

**Fig 5. PAM1.4 Fab interaction analysis to DBL2.** Similar to Fig 4, Binding analysis of PAM1.4 Fab to DBL2 is represented using Ligplot+ and UCSF ChimeraX. FCR3 VAR2CSA sequence positions 905 and 958 corresponds to alignment positions 982 and 1073 respectively.

immunity to placental malaria [36]. However, the relative importance of neutralization and opsonization remains unknown. Indeed, acquired antibodies to VAR2CSA are mainly cytophilic [37] and importantly are markedly afucosylated [18]. This trait is shared with IgG elicited by other host membrane-associated proteins such as antigens of enveloped virus [38]. Thus, Fc-modulated versions of PAM1.4 and other broadly reactive but non-inhibitory antibodies may serve as useful tools for benchmarking vaccine-elicited cell-mediated immunity to placental malaria.

The structural resolution of VAR2CSA has raised hopes that novel immunogen and vaccine strategies for boosting cross-reactive responses can be designed [39]. However, for this to succeed, identification and characterization of new broadly reactive, and ideally, CS-binding inhibitory VAR2CSA antibodies are likely needed. Negative stain electron microscopy for polyclonal epitope mapping (nsEMPEM), is a novel method, which can help dissecting naturally acquired or vaccine-induced immunity. This technique has been developed to discern the polyclonal responses in non-human primates to vaccination with HIV immunogens and was used to assess the polyclonal human antibody response after vaccination against or infection with COVID-19 [40, 41]. Here we have shown that VAR2CSA is a good candidate for nsEMPEM by mapping epitope binding region of PAM1.4 Fab based on negative stain (S2 Fig). Thus, nsEMPEM studies of naturally acquired and vaccine-induced antibody epitopes are now possible and could enhance the development and assessment of improved second-generation VAR2CSA vaccines.

## Methods

### Ethics statement

Ethical clearance was granted by the Medical Research Coordinating Committee of the National Institute for Medical Research (reference number NIMR/HQ/R.8a/Vol. IX/1717).

### Recombinant protein production

VAR2CSA full length was produced in the baculo virus—insect cell expression system based on previously described method [13]. Gene fragment from N terminal M1 to F2649 of DBL6 was cloned into baculovirus vector pAcGP67-A (BD Biosciences) with V5 and 6xHisTag on the C terminal. The construct was transfected into Sf9 cells (Spodoptera frugiperda-9) to generate and amplify the viral particles which were then used to infect High-Five (X5) insect cells. Expression was induced for 48h-60h. Cells were then pelleted by centrifugation at 10000xg for 15mins at 4°C and discarded. The media supernatant was concentrated, and buffer exchanged to PBS pH. 7.4 buffer using 50k MWCO hollow-fiber filter. Imidazole was added to a final concentration of 60mM and loaded onto 5ml HisTrap HP column (Cytiva). Protein was eluted through a linear gradient of PBS with 300mM Imidazole pH. 7.4 buffer. The protein was further purified by subjecting to size exclusion chromatography using Superose 6 Increase 10/300 column (Cytiva) in 20mM Tris-HCl, 150mM KCl, 0.05% Sodium Azide, pH. 7.5. Protein was concentrated to 1.4mg/ml using vivaspin (sartorius), 100kDA MWCO, aliquoted and stored at -80°C for further use.

VAR2CSA fragments DBL1-ID2a, ID1-ID2a, DBL4 and DBL5 were produced using E. coli Shuffle cells (NEB). Briefly, cells were harvested and homogenized after IPTG induction. Expressed protein from the cytoplasmic fraction was purified using HisTrap HP column (Cytiva) followed by a cation exchange polishing step using HiTrap SP HP (Cytiva). Purity of the VAR2CSA fragments was assessed by SDS-PAGE (S10 Fig). Folding of the proteins was assessed by confirming CSPG binding and by reactivity of serum from placental malaria-

exposed Tanzanian [42] and Ghanaian (N = 20) women [43, 44] and Ghanaian male adults (S10 Fig).

## PAM1.4 IgG Expression and purification

The human monoclonal IgG1 antibody PAM1.4 has been previously characterized [26]. In this study, PAM1.4 was produced as recombinant secreted protein in suspension-adapted Human Embryonic Kidney cells (Expi293F Gibco), as described in [45]. In brief, plasmids encoding the constant regions of the γ1-chain and of the κ-chain plus the variable domains of the PAM1.4 antibody were co-transfected into Expi293F cells using the Expi293 Expression System (Gibco), following manufacturer's instructions. The secreted antibody was harvested six days post transfection, clarified by centrifugation and filtration, and subsequently purified using protein G plus agarose columns (Pierce). The antibody was eluted in 0.1 M glycine pH. 2.8 and immediately neutralized with Trizma hydrochloride solution 1 M pH 9.0. The PAM1.4 IgG was buffer exchanged into PBS using Amicon ultra centrifugal concentrators (Millipore) and quantified by NanoDrop2000 (Thermo Scientific).

## PAM1.4 Fab production and purification

To produce PAM1.4 Fab, PAM1.4 IgG was buffer exchanged into PBS, 20mM L-cysteine HCl, pH. 7.4. by 30kDa MWCO membrane centrifugal filter units (Cytiva). Fabs were generated by papain digestion using immobilized papain agarose beads (Sigma-aldrich) in PBS, 20mM L-cysteine-HCl, 2mM EDTA, pH. 7.4 for 120 mins at 37˚C in a 1ml spin column. Papain beads were removed by centrifugation @ 1000xg for 2 mins, flow through was collected and subjected to Protein A spin column incubated 30 mins to remove undigested IgGs and Fc fragments. Flow through containing Fabs was collected, further purified by size exclusion chromatography using Superdex 200 increase 10/300 column (Cytiva) in TBS, pH. 7.5, before concentrating and storage at 4˚C.

## Enzyme Linked Immuno Sorbent Assay–ELISA

A Falcon 96-well clear flat bottom microplate was coated with PAM1.4 Fab-fragment, decorin (Sigma-Aldrich) or VAR2CSA full-length and individual domains in PBS (3μg/ml) overnight at 4˚C. After incubating the plate was blocked for 1h at RT with PBS-T (PBS, 0.05% (v/v) tween-20, 1% BSA (w/v), pH. 7.4). Full-length VAR2CSA and shorter constructs (ID1-ID2a, DBL1-ID2a, DBL4 and DBL5) were added in a 2-fold dilution series in PBS from 50nM-1.56nM. Serum binding was assessed by adding diluted serum in PBS-T (1:200, 1:400, 1:800, and 1:1600) to plates coated with VAR2CSA. After incubation for 1h, plates were washed and added anti-HIS HRP (Miltenyi) diluted 1:3000 for VAR2CSA detection or anti-human Fc (1:5000) for IgG in serum detection. The plates were developed using TMB-plus2 (kementec) and the absorbance at 450nm was read.

## Immunomagnetic selection of P. falciparum-infected erythrocytes

Erythrocytes infected by late-stage IT4/FCR3, NF54, and 7G8 parasites were selected for surface expression of VAR2CSA using protein A-coupled DynaBeads coated with PAM1.4 IgG as described previously [46]. Surface expression of the corresponding PfEMP1 protein was monitored by flow cytometry [46]. Intact and unfixed late-stage infected erythrocytes were acquired on a Beckman Coulter FC500 flow cytometer and data analyzed with FlowLogic software (version8.3; Inivai Technologies, Australia). The parasite genotypes and the absence of *Myco-plasma* contamination (MycoAlert Mycoplasma Detection Kit; Lonza) were verified regularly.

## Reactivity to native VAR2CSA on the infected erythrocyte surface

PAM1.4 IgG reactivity to native VAR2CSA expressed on the infected erythrocyte surface was detected by flow cytometry as previously described [46]. Briefly, late-stage infected erythrocytes purified by MACS were labelled with PAM1.4 (0.16 to 10 µg/mL) followed by FITC-conjugated anti-human IgG (1:150; Vector Laboratories). Parasite nuclei were stained with ethidium bromide (2 µg/mL). Intact and unfixed infected erythrocytes were acquired by flow cytometry (Beckman Coulter FC500) and data analyzed using FlowLogic software (Inivai Technologies, Australia).

## CS-specific adhesion of infected erythrocytes

Static adhesion assays were employed to evaluate the capacity of human antibodies to inhibit the infected erythrocyte adhesion to CS, as described elsewhere [47, 48]. The PAM1.4 Fab fragment, two preparations of total IgG purified from pools of ten plasma samples (non-pregnant Ghanaian women with natural *P. falciparum* exposure) with high and low levels of anti-VAR2CSA IgG, respectively, and purified non-immune human IgG (Sigma-Aldrich) were tested. Petri dishes were coated overnight with PBS containing 5 µg/mL decorin (chondroitin sulfate proteoglycan [CSPG]; Sigma-Aldrich). After blocking, a 20% parasite suspension of purified late-stage IT4VAR04 (aka FCR3 VAR2CSA) infected erythrocytes was preincubated with the antibodies (100 µg/mL) or soluble CSA (500 µg/mL) and allowed to adhere to petri dishes. Non-adhering erythrocytes were removed, and remaining bound cells fixed with 1.5% glutaraldehyde in PBS and stained with 10% Giemsa. Adhering cells were quantified by light microscopy as the number of infected erythrocytes bound/mm$^2$ and compared with the level of binding in the absence of antibody (PBS, control for maximum binding).

## Cryo-EM sample preparation

Full-length VAR2CSA was complexed with PAM1.4 Fab by mixing at 1:1.5 molar ratio (VAR2CSA: PAM1.4 Fab) and the final concentration of VAR2CSA was 0.8mg/ml. Quantifoil 1.2/1.3 carbon grids were glow-discharged with Leica Coater ACE200 for 60s at 5mA current. Cryo-EM grids were prepared with Vitrobot Mark IV (Thermo Fisher) operated at 100% humidity and 4˚C. 3µl of the complexed sample was applied to the grids, incubated 3s, blotted for 3s and plunge frozen in liquid ethane and then stored in liquid nitrogen for data collection [13].

## Cryo-EM data collection and processing

Full-length VAR2CSA in complex with PAM1.4 Fab micrographs were collected on Titan Krios electron microscope (FEI)– 300kV. 4927 movies were obtained on Falcon3 direct electron detector operating in counting mode at a pixel size of 0.832Å and a total dose of 44 e/Å$^2$ over 39 frames using a defocus range -1 to -2.8µm. 4927 movies were patch motion corrected for beam induced motion within cryoSPARC v3.31 [49]. The non-dose weighted movies were used to estimate CTF parameters using CTFFIND4 [50] plugged in cryoSPARC v3.31. Blob particle picking was done to all micrographs and particles were extracted to perform 2D classification. Template based particle selection was done to re-extract particles from micrographs by Local Motion Correction with dose-weighting at a box size of 440 pixels. An initial 925142 particle stack was extracted and subjected to iterative rounds of reference-free 2D classifications to identify class averages corresponding to FL VAR2CSA structural features. The initial Ab-initio classification, followed by heterogeneous refinement provided one good class with well- defined VAR2CSA structural characters with 414806 particles. These particles were

further 3D classified and heterogeneously refined to reveal two distinct volumes, one is APO VAR2CSA without PAM1.4 Fab (207129 particles) and another, a complex of VAR2CSA and PAM1.4 Fab (165911 particles). Particles from APO VAR2CSA and PAM1.4 –VAR2CSA complexes were separately refined using non-uniform 3D refinement imposing C1 symmetry. APO VAR2CSA reached 3.12Å, and PAM1.4 –VAR2CSA complex reached 3.04Å final resolution according to the gold-standard FSC [51].

## Cryo-EM structure modelling and refinement

Initial coordinates of VAR2CSA and PAM1.4 Fab were generated by docking individual chains of the reference structures into the density map in UCSF Chimera [52] followed by molecular dynamics flexible fitting (MDFF) [53]. For APO VAR2CSA, the model co-ordinate for VAR2CSA used was PDB ID: 7B52. For PAM1.4 Fab, initial model was generated by alphafold2-multimer [54]. Heavy and light chain sequences were given as single input and the top ranked model generated the mean confidence score, pLDDT>96. The model building of APO VAR2CSA and VAR2CSA–PAM1.4 Fab was done iteratively using COOT [55] followed by real space refinement in PHENIX package [56]. Local resolution of the maps (APO VAR2CSA and PAM1.4 Fab–VAR2CSA) were calculated in cryoSPARC v3.31. Validation of model coordinates was performed in MolProbity [57] and is reported in S1 Table.

## Epitope characterization and sequence analysis

Structure figures were generated using PyMOL (Version 2.1 Schrodinger, LLC) or UCSF ChimeraX [52, 58, 59]. The interacting residues and the figures for hydrogen bonds, electrostatic and hydrophobic interactions were generated using Ligplot+ [60]. Epitope conservation analysis of VAR2CSA residues were performed using previously aligned 3737 VAR2CSA sequences and visualized using WebLogo 3 [13, 61]. LOGO plots were generated based on 1388 sequences of NTS-DBL6 extracted from the var database, varDB PF3K [62]. Antibody sequence analysis was performed using the IMGT (International ImMunoGeneTics information system)/V-QUEST (V-QUEry and STandardization) tool and database [63].

## Negative stain electron microscopy

Purified APO VAR2CSA and VAR2CSA PAM1.4 Fab complex at a concentration of 20μg/ml were adsorbed on a glow discharged PureC 300 mesh carbon-coated copper grids (EMD Science) for 1 min followed by 2% Uranyl Acetate (Electron Microscopy Sciences) staining. Method was adopted from [64] and the data collection parameters are detailed in S2 Table. Raw Micrographs were recorded in TALOS Arctica 200kV (Thermo Fisher Scientific) with Falcon III direct electron detector (Thermo Fisher Scientific) at room temperature using SerialEM 3.7 [65]. Images were collected in integrating mode at a pixel size of 1.54 Å/px, using flux of 1.8 e/Å$^2$/sec and a total dose of 43.5 e/Å2 over a total of 24 frames, with defocus values ranging from -0.6 to -1.2μm at 0.2μm intervals. Images were processed in cryoSPARC v3.31, and a reference free particle stack was created by gaussian blob picker [49]. Particles corresponding to PAM1.4 Fab complexed with VAR2CSA were identified by multiple rounds of 2D classification to identify classes displaying structural elements of Fabs on VAR2CSA. Particles were extracted to generate ab initio models in cryoSPARC v3.31 and further processed to segregate APO VAR2CSA and PAM1.4 complex structures. FSC (0.5) of VAR2CSA PAM1.4 Fab map was calculated to be 22Å (S11 Fig). Figures were prepared in UCSF Chimera [52, 59].

## Supporting information

**S1 Fig. Reactivity of PAM1.4 to native VAR2CSA expressed on the surface of infected erythrocytes.** (a) Representative flow cytometry histogram of FCR3 VAR2CSA in the presence of 10µg/ml PAM1.4 IgG (cyan) and 10µg/ml IgG isotype control (gray). (b,c) Erythrocytes infected by FCR3 VAR2CSA were incubated with PAM1.4 (seven two-fold serial dilutions from 0.16 to 10µg/ml). The percentage of FITC-positive cells (infected erythrocytes binding PAM1.4 or IgG isotype control) and the median fluorescence intensity were estimated and used to create dose-response curves.
(PDF)

**S2 Fig. Model fitting of APO VAR2CSA (PDB ID: 7B52) to map the epitope region of PAM 1.4 Fab as a proof of concept technique to show VAR2CSA Full length is an ideal target for negative stain based polyclonal epitope mapping.** (a) APO VAR2CSA structure is modelled in the negative stain density of VAR2CSA PAM 1.4 Fab complex to identify and split the density corresponding to the Fab (b) Potential epitopes mapped based on the surface area of the Fab interacting with APO VAR2CSA model structure (c) CryoEM structure of VAR2CSA PAM 1.4 Fab overlayed to verify the density fit and epitopes mapped.
(PDF)

**S3 Fig. CryoEM workflow on determining VAR2CSA PAM 1.4 Fab structure.** (a) Representative micrograph of the CryoEM grid used for structure determination. (b) Representative 2D class averages with box size of 44nm. (c) Flow chart of the data processing. Detailed method flow is described in the method section and S1 Table. (d) Final map of VAR2CSA PAM 1.4 Fab complex colored based on calculated resolution. Pam 1.4 Fab is magnified to highlight the resolution at the binding region of the Fab (Paratope). (e) Final map of APO VAR2CSA colored based on resolution. (f,g) Gold Standard Fourier Shell Correlation (FSC) curve of VAR2CSA PAM 1.4 (3.04 Å) and APO VAR2CSA (3.12 Å) respectively.
(PDF)

**S4 Fig. Different set of views of VAR2CSA + PAM1.4 Fab cryoEM density map and its corresponding refined ribbon model.**
(PDF)

**S5 Fig. Molecular architecture of APO-VAR2CSA structure resolved from VAR2CSA unbound to PAM1.4 Fab particles and flipped 180˚ to the right.** Domains are color coded as depicted above.
(PDF)

**S6 Fig. PAM1.4 does not inhibit the infected erythrocytes adhesion to CSA.** Percentage of FCR3 VAR2CSA infected erythrocytes binding to CSA in the presence of 100 µg/ml antibodies or 500 µg/ml soluble CSA (sCSA). Control, absence of antibodies; Fab, PAM1.4 Fab fragment; PAM1.4, PAM1.4 whole IgG; high and low, corresponds to total IgG purified from a pool of plasma with high and low levels of anti-VAR2CSA, respectively; IgG, IgG isotype control. Median values ± 95% CI from two independent experiments and P values using Kruskal-Wallis test followed by Dunn's multiple comparisons test are shown.
(PDF)

**S7 Fig. Quality of the cryoEM density.** (a) ID1 loop: N505 –K522; DBL2: L905 –N908, K914 –W917, Y958 –A963; DBL4: E1614 –R1617, K1872 –E1875 (b) Heavy chain CDR loops, CDR–H1: D47 –G52; CDR–H2: I70 –K77; CDR–H3: R117 –N131. (c) Light chain CDR loops, CDR–L1: E46 –N50; CDR–L2: I67 –A70; CDR–L3: Q108 –A116. The volume surface is zoned

at 2.5Å in UCSF ChimeraX for all the regions.
(PDF)

**S8 Fig. Epitopes of PAM 1.4 Fab interacting by hydrophobic contacts.** Binding plot analysis is done using Ligplot+ v.2.7 as shown in Figs 4 and 5. Hydrophobic interaction is shown as red spokes radiating from the residues involved in the interaction. DBL2 and DBL4 are involed in binding with CDR–L2, H2 and H3. Non-CDR residue P94 of heavy chain interacts with ID2 residue I1055. Weblogo is used to represent sequence conservation of the epitopes.
(PDF)

**S9 Fig. PAM1.4 sequence annotation.** Alignment and annotation of frame work (FR) and complementarity-determining regions modified from IMTG/quest output. Green letters indicate amino acids with markedly different biochemical properties than the germline sequence. Red letters indicate residues forming hydrogen bonds or electrostatic interactions with VAR2CSA. Bold underlined letters indicate residues stabilizing VAR2CSA binding through hydrophobic interactions.
(PDF)

**S10 Fig. Quality control of individual domain proteins of VAR2CSA.** (A) SDS representation of domain constructs both reduced and non-reduced bands corresponding to the molecular weight. DBL5 (37kDa), DBL4 (53kDa), ID1-ID2a (68kDa), DBL1-ID2a (110kDa). (B-D) ELISA binding of full-length VAR2CSA and individual domains by pools of serum derived from Tanzanian (B) and Ghanaian (C) placental malaria-exposed women and Ghanaian adult males (D), in a dilution series 1:200, 1:400, 1:800 and 1:1600. (E) Decorin CSPG binding assessed by ELISA. Only DBL1-ID2a and ID1-ID2a are showing interaction, as these constructs contain the CSA binding site.
(PDF)

**S11 Fig. GSFSC curve of VAR2CSA PAM 1.4 Fab negative stain map.** The resolution calculated at 0.5 FSC is 22Å.
(PDF)

**S1 Table. CryoEM data collection and refinement statistics for VAR2PAM 1.4.**
(PDF)

**S2 Table. Negative stain data collection parameters for VAR2PAM 1.4.**
(PDF)

## Acknowledgments

SSRR and KTW acknowledge access to computational resources from the Danish National Supercomputer for Life Sciences (Computerome). We would very much like to acknowledge Core Facility for Integrated Microscopy (CFIM), University of Copenhagen for providing support during data collection. We also would like to acknowledge Swedish National Cryo-EM facility, Science for Life laboratory, Sweden for support on negative stain data collection.

## Author Contributions

**Conceptualization:** Sai Sundar Rajan Raghavan, Morten Agertoug Nielsen, Lea Barfod, Lars Hviid, Ali Salanti, Thomas Lavstsen, Kaituo Wang.

**Data curation:** Sai Sundar Rajan Raghavan, Kaituo Wang.

**Formal analysis:** Sai Sundar Rajan Raghavan, Robert Dagil, Mary Lopez-Perez, Swati Choudhary, Tobias Gustavsson, Yong Wang.

**Funding acquisition:** Pontus Gourdon, Morten Agertoug Nielsen, Lea Barfod, Lars Hviid, Ali Salanti, Thomas Lavstsen.

**Investigation:** Sai Sundar Rajan Raghavan, Robert Dagil, Mary Lopez-Perez, Sebastian Boje Christensen.

**Methodology:** Sai Sundar Rajan Raghavan, Robert Dagil, Mary Lopez-Perez, Julian Conrad.

**Resources:** Maria Rosaria Bassi, Maria del Pilar Quintana, Michael Fokuo Ofori, Daniel Thomas Remias Minja, Christentze Schmiegelow.

**Visualization:** Sai Sundar Rajan Raghavan, Robert Dagil, Mary Lopez-Perez.

**Writing – original draft:** Sai Sundar Rajan Raghavan, Robert Dagil, Thomas Lavstsen, Kaituo Wang.

**Writing – review & editing:** Sai Sundar Rajan Raghavan, Robert Dagil, Mary Lopez-Perez, Maria Rosaria Bassi, Maria del Pilar Quintana, Swati Choudhary, Tobias Gustavsson, Yong Wang, Pontus Gourdon, Michael Fokuo Ofori, Sebastian Boje Christensen, Daniel Thomas Remias Minja, Christentze Schmiegelow, Morten Agertoug Nielsen, Lea Barfod, Lars Hviid, Ali Salanti, Thomas Lavstsen, Kaituo Wang.

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
