## [Decision Letter · Decision Letter 0]

29 Aug 2022

Dear Dr. wang,

Thank you very much for submitting your manuscript "Cryo-EM reveals the conformational epitope of human monoclonal antibody PAM1.4 broadly reacting with polymorphic malarial protein VAR2CSA" for consideration at PLOS Pathogens. As with all papers reviewed by the journal, your manuscript was reviewed by members of the editorial board and by several independent reviewers. In light of the reviews (below this email), we would like to invite the resubmission of a significantly-revised version that takes into account the reviewers' comments.

The reviewers have raised a number of issues that need to be addressed before your paper would be considered acceptable for publication in PLOS Pathogens. In particular, please consider their comments on the need for additional experimental controls, and more clearly address the potential protective functions of the MAb. Please comment on the ways in which these findings can inform vaccine development, as noted by Reviewer 2, since the MAb does not have the desired property of inhibition of binding to CS.

We cannot make any decision about publication until we have seen the revised manuscript and your response to the reviewers' comments. Your revised manuscript is also likely to be sent to reviewers for further evaluation.

Sincerely,

James G. Beeson, MBBS, PhD

Associate Editor

PLOS Pathogens

Kami Kim

Section Editor

PLOS Pathogens

Kasturi Haldar

Editor-in-Chief

PLOS Pathogens

orcid.org/0000-0001-5065-158X

Michael Malim

Editor-in-Chief

PLOS Pathogens

orcid.org/0000-0002-7699-2064

The reviewers have raised a number of issues that need to be addressed before your paper would be considered acceptable for publication in PLOS Pathogens. In particular, please consider their comments on the need for additional experimental controls, and more clearly address the potential protective functions of the MAb. Please comment on the ways in which these findings can inform vaccine development, as noted by Reviewer 2

Reviewer's Responses to Questions

**Part I - Summary**

Reviewer #1: Understanding the molecular and structural basis for broadly inhibitory antibodies to VAR2CSA would be critical in the development of a broadly protective placental malaria vaccine. Here the authors use cryo-EM to evaluate the binding site of the broadly reactive antibody, PAM1.4 Fab, and found that it binds a highly conserved conformational epitope involving several domains of the core VAR2CSA structure. The authors suggest that while broadly reactive, that this antibody does not functionally block the interaction of VAR2CSA and CS.

The study employs a useful technique for resolving protein structures – cryo-EM – to generate molecular details of the interaction of PAM 1.4. This data is compelling and adds additional details to the structure-based approaches for VAR2CSA vaccine target design, in addition to further validating the utility of negative stain and cryo-EM approaches for resolving protein structures in malaria, and specifically for VAR2CSA. The structural data is a strength of the paper. Some of the binding and functional experiments could be further strengthened and appropriate controls included so that the conclusions are truly justified by the data presented.

The paper is clear and well-written. I hope that the suggestions below will help the authors improve the manuscript.

Reviewer #2: This study has resolved the cryoEM structure of full length VAR2CSA ectodomain in complex with a broadly neutralizing antibody (Fab of PAM 1.4). The work is for the most part presented clearly and carefully performed. While the structure reveals the epitope of PAM 1.4, the study lacks defining a mechanism of action of the monoclonal antibody. This is further detailed in the "Major Issues" section of the review. Overall this is an interesting structural study and with some revisions could be further strengthened.

**Part II – Major Issues: Key Experiments Required for Acceptance**

Reviewer #1: Results:

1. Fig 1A. Line 91-93: Validation of the expressed VAR2CSA domains would be important especially when showing no direct binding. Can positive binding be demonstrated with polyclonal polygravid immune serum or vaccinated IgG? This would be important to demonstrate the protein domains are properly folded. This will be especially important to demonstrate given the conclusions stated in Lines 114-115. To make such claims, the authors should demonstrate that the protein domains themselves are properly folded and able to be bound by antibodies previously shown to react to these domains. This will make the contrast w with PAM 1.4 more dramatic and convincing.

2. Fig 1B. Line 93-95: Can the authors verify that there is strain specific binding by flow of the other monoclonal antibodies? (Thus PAM1.4 Fab, in contrast to other antibodies, is truly pan-strain binding). A negative control condition (for example, non-CSA panned parasites of the same strains that should show decreased VAR2CSA expression and decreased binding, or DTT and Trypsin treatment to allow cleavage of VAR2CSA from the surface of the IEs), should be included to show that the positive binding is clearly specific and reversible. The flow-through from the mAb-bead-column preparations would also be a helpful control. Here, the authors used beads coupled with the mAb PAM1.4 to pre-select for PAM1.4 binding late trophozoites and schizonts. It would be helpful to see what percentage of VAR2CSA expressing IEs are truly recognized by PAM 1.4.

3. Figure S1. It would be ideal if the authors could provide flow data using an isotype matched unrelated antibody as the negative control to rule out non-specific binding.

4. Figure S7. A major finding of the paper, and a point of emphasis of the authors in the introduction and discussion, this data should be moved to the main text, but would require additional replicates. The error bars are a bit wide for the two replicates and appears that PAM1.4 inhibits more significantly in one experiment compared to the second. The authors should describe the characteristics of each antibody in the legend: Fab = PAM 1.4 Fab and IgG = PAM 1.4 total IgG?

Reviewer #2: What is shown is that the antibody binds some conserved residues (up to 3/4 of VAR2CSA variants) that are distant from the binding site with CSA. The failure of PAM 1.4 to inhibit VAR2CSA binding to CSA is also shown (Figure S7). It would have been good for the authors to further examine mechanism. Does the PAM 1.4 IgG interact with complement, monocytes or Fc-gamma-Rs to carry out a broadly protective antibody-mediated activity? Or does PAM 1.4 IgG just bind a broad range of VAR2CSA variants without any correlates of protective immunity. These data are needed for the various claims in the manuscript around how this structural work could benefit future vaccine design. Should the above data not be available then the tone of the manuscript needs to be modified throughout. Some minor comments are also provided below.

**Part III – Minor Issues: Editorial and Data Presentation Modifications**

Reviewer #1: Suggestions:

1. The manuscript represents a very interesting finding and an important use of cryo-EM to help resolve structural interactions between an interesting mAb and its binding domain on VAR2CSA. However, the manuscript feels brief (almost as if it has been written in the format of a short report). Some of the additional data presented in supplemental (for example S3, S4, S5, S6, S7) could be moved to the main text as panels within the existing figures to provide more detail and clarity.

Clarifications:

2. Line 81-82 – The authors claim “whereas the early data indicating that PAM1.4 was also able to inhibit IE adhesion to CS have not been subsequently verified.” Please cite the source of this early data (reference if published) and indicate why there is controversy as to whether this data is credible, or in need of further verification. Do the authors mean that it has never been reproduced or independently verified? In the discussion (Lines 180-181), the authors mention that there is unpublished data from several groups. Can these data be listed as “personal communication” or “unpublished” and attributed to the groups/authors?

3. Lines 142-147: Presented under results, it is unclear if the authors are referring to their own alignments or merely citing previously published studies of VAR2CSA diversity. It seems that the later is the case, which is fine, but if so this should be moved into the discussion. If the authors have generated sequence alignments themselves to look at diversity of the PAM 1.4 binding epitopes (perhaps of the specific strains they used in their binding assays), this data should be presented in the paper as a figure allowing the reader to visualize the highly conserved versus slighly more variable epitopes included in the mAb binding region.

4. Lines 205-209: The authors’ final conclusion is that PAM 1.4 is broadly reactive, but not neutralizing. However, in the introduction, the authors mention that PAM1.4 can be protective via the mechanism of opsonization of IEs for Fc receptor-dependent killing in vitro, citing previous studies. The authors should comment on the relative contributions of these two mechanisms in the context of protection from placental malaria in the discussion.

Some suggestions for typos/grammar:

5. Line 55 – “VAR2CA variants” - VAR2CSA variants

6. Throughout: even though the first mention of a term is correctly followed by an abbreviation, there are some abbreviations that feel unnecessary and are a little distracting. The authors might consider leaving the unabbreviated term, especially for shorter phrases (such as IEs – infected erythrocytes; PM – placental malaria, etc). This is merely a stylistic suggestion.

Reviewer #2: Abstract, "High-resolution data shows..." the structure was determined at a global resolution of around 3Å and at around 2.5 Å at the antibody:antigen interface. This does not represent high resolution, but actually medium to low resolution data. Please modify here and and other references to high resolution in the manuscript.

Lines 94 - 96 (Fig 1B and S1) describes flow cytometry of PAM1.4 binding to VAR2CSA variant infected RBCs. However, there is no suitable negative controls provided (a non-VAR2CSA parasite; isotype control - or at least this is not shown in the figure captions).

Fig 1B. These ELISA plots are single lines without information about variation. How many replicates were performed and if sufficient repeats then appropriate errors should be included. If only duplicates it is best practice to plot both data points against the lines of best fit.

Fig 2 is fine to give an overall representation of the cryoEM structure. It would be good to also include here or in a separate figure a cleaner set of views of the refined structural model.

Fig 3 shows interactions but much of the details are hard to see. In particular text sizes are particularly small across this entire figure.

PLOS authors have the option to publish the peer review history of their article (what does this mean?). If published, this will include your full peer review and any attached files.

Reviewer #1: No

Reviewer #2: No
---

## [Decision Letter · Decision Letter 1]

10 Oct 2022

Dear Dr. wang,

We are pleased to inform you that your manuscript 'Cryo-EM reveals the conformational epitope of human monoclonal antibody PAM1.4 broadly reacting with polymorphic malarial protein VAR2CSA' has been provisionally accepted for publication in PLOS Pathogens.

Best regards,

James G. Beeson, MBBS, PhD

Associate Editor

PLOS Pathogens

Kami Kim

Section Editor

PLOS Pathogens

Kasturi Haldar

Editor-in-Chief

PLOS Pathogens

orcid.org/0000-0001-5065-158X

Michael Malim

Editor-in-Chief

PLOS Pathogens

orcid.org/0000-0002-7699-2064

The authors have appropriately addressed the comments and requests from the reviewers.

Reviewer Comments (if any, and for reference):

Reviewer's Responses to Questions

**Part I - Summary**

Reviewer #2: This study provides new insight into the interaction between a candidate monoclonal antibody and the main target of pregnancy-associated malaria, VAR2CSA. The cryoEM structure has been carefully determined and the analysis of the structural data is interesting. Some additional experiments have been provided to demonstrate the antibody specifically binds to its target and broadly cross-reacts with several common variants of VAR2CSA. A discussion of the monoclonal antibody mechanism of action is supported by previous publications and is relatively sound and interesting. Altogether, this work provides new insights relevant to the development of a monoclonal antibody candidate that may extend to vaccine development, although the path for using this information for effective vaccine design is a little less tangible. Overall this work is of general interest to the malaria immunity community and could provide support for future immunotherapies.

**Part II – Major Issues: Key Experiments Required for Acceptance**

Reviewer #2: In their revisions the authors have addressed or adequately responded and revised their manuscript to address my original concerns. I have no further major or minor revisions to suggest.

**Part III – Minor Issues: Editorial and Data Presentation Modifications**

Reviewer #2: No additional comments.

PLOS authors have the option to publish the peer review history of their article (what does this mean?). If published, this will include your full peer review and any attached files.

Reviewer #2: No

---

## [Editor Report · Acceptance letter]

2 Nov 2022

Dear Dr. wang,

We are delighted to inform you that your manuscript, "Cryo-EM reveals the conformational epitope of human monoclonal antibody PAM1.4 broadly reacting with polymorphic malarial protein VAR2CSA," has been formally accepted for publication in PLOS Pathogens.

Best regards,

Kasturi Haldar

Editor-in-Chief

PLOS Pathogens

orcid.org/0000-0001-5065-158X

Michael Malim

Editor-in-Chief

PLOS Pathogens

orcid.org/0000-0002-7699-2064